# Novel Cognitions in Allelopathy: Implications from the “Horizontal Natural Product Transfer”

**DOI:** 10.3390/plants11233264

**Published:** 2022-11-28

**Authors:** Laura Lewerenz, Sara Abouzeid, Mahdi Yahyazadeh, Tahani Hijazin, Dirk Selmar

**Affiliations:** 1Institut für Pflanzenbiologie, Technische Universität Braunschweig, Mendelssohnstraße 4, D-38106 Braunschweig, Germany; 2Pharmacognosy Department, Faculty of Pharmacy, Mansoura University, Mansoura 35516, Egypt; 3Research Institute of Forests and Rangelands, Agricultural Research, Education and Extension Organization (AREEO), Tehran P.O. Box 13185-116, Iran; 4Biology Department, Faculty of Science, Mutah University, P.O. Box 7, Mutah 61710, Jordan

**Keywords:** horizontal natural product transfer, allelopathy, alkaloids, uptake

## Abstract

Whereas the translocation of allelochemicals between plants is well established, a related general transfer of genuine specialized metabolites has not been considered so far. The elucidation of the so-called “Horizontal Natural Product Transfer” revealed that alkaloids, such as nicotine and pyrrolizidine alkaloids, which are leached out from decomposing alkaloid-containing plants (donor plants), are indeed taken up by the roots of plants growing in the vicinity (acceptor plants). Further studies demonstrated that phenolic compounds, such as coumarins or stilbenes, are also taken up by acceptor plants. Contemporary analyses from co-cultivation experiments outlined that natural products are not exclusively transferred from dead and rotting donor plant materials, but also from vital plants. In analogy to xenobiotics, the imported specialized metabolites might also be modified within the acceptor plants. As known from the uptake of xenobiotics, the import of specialized metabolites is also generally due to a simple diffusion of the substances across the biomembranes and does not require a carrier. The uptake depends in stricto sensu on the physicochemical properties of the certain compound. This article presents a current overview of the phenomenon of “Horizontal Natural Product Transfer” and discusses its relevance for our understanding of allelopathic interactions. The knowledge that specialized metabolites might in general be readily translocated from one plant into others should significantly contribute to our understanding of plant–plant interactions and—in particular—to the evolution of typical allelopathic effects, such as inhibition of growth and germination of potential competitors.

## 1. Background

By their roots, plants take up plenty of inorganic nutrients from the soil, as well as many other compounds. Concerning biochemical ecology, in particular, the uptake of allelochemicals, i.e., active substances, which inhibit germination or growth of putative competitors, is of special interest [1]. In general, these substances are released from donor plants into the soil and exhibit their inhibitory effect on the plants growing in their vicinity [2,3] (see also other chapters of this Special Issue). To execute their inhibitory effects, the allelochemicals have to be taken up by the acceptor plants [1,4]. A further import of natural products is related to Systemic Acquired Resistance (SAR) of plants: it was shown that salicylic acid, which provokes SAR, is also taken up by roots [5,6]. In this context, it has to be emphasized that the uptake of ionic compounds such as salicylic acid, as well as of nutrients such as nitrate, sulfate or metal ions, requires specific transporters [7,8,9]. As a consequence, by regulating the carrier proteins’ number and activity, plants are capable of controlling the import of the related substances by their roots. As opposed to this, the situation is entirely different when considering the uptake of xenobiotics, i.e., systemic herbicides, fungicides [10], or veterinary medicines [11]. Because of their partially hydrophobic character, these substances are able to diffuse smoothly through membranes [12,13,14]. Thus, these substances are taken up by the roots passively (see below), and, in consequence, the plants cannot control or even prevent the import of these substances. Although these nexuses were well established, corresponding contemplations about common specialized metabolites have not been considered at all. However, this attitude has plainly changed within the last few years due to comprehensive investigations to identify the potential sources of various contaminations of plant-derived commodities.

Triggered by alarming reports of the European Food Safety Authority (EFSA) on contaminations of herbal medicines and phytopharmaceuticals by various toxic alkaloids, i.e., nicotine [15], pyrrolizidine alkaloids [16,17], or tropane alkaloids [18], several studies on the source of these contaminations were conducted. It became apparent that plants take up nicotine [19] and pyrrolizidine alkaloids [20] from the soil. Subsequent field trials demonstrated that nicotine, which is leached out from discarded cigarette butts, is taken up massively by acceptor plants. The related nicotine concentrations in the commodities were around 1 µg/g d.w. [21], thus exceeding the official maximum residue limit (0.01 µg/g d.w.) about a hundred-fold. In the same manner, pyrrolizidine alkaloids (PAs), which are leached out from rotting PA-containing weeds into the soil, e.g., *Senecio jacobaea*, are also taken up by the roots of the acceptor plants and translocated into their leaves [20,22]. Accordingly, such transfer of alkaloids—at least in part—accounts for the numerous and widespread alkaloidal contaminations of spice and medicinal plants reported comprehensively [15,16,23]. Based on these coherences, the so-called “Horizontal Natural Product Transfer” was outlined [24], a so far unrecognized phenomenon which has been studied intensively during the last few years (for review, see [25,26,27]). It became evident that apart from alkaloids, other natural products, such as coumarins [28], stilbenes [29], or betalains [30], are also taken up by plant roots. In this context, it has to be mentioned that in some cases, the natural products whose import into acceptor plants was verified, i.e., the stilbene resveratrol [29] or the coumarins [27], are known to exhibit allelopathic effects [31,32]. These examples vividly demonstrate that the line between “simple natural products” and “allelochemicals”, which are leached out, is quite diffuse. Up to now, this issue has not been taken into consideration when evolutionary aspects of allelopathic effects are discussed.

## 2. Horizontal Natural Product Transfer—Basic Principles and Cognitions

### 2.1. Uptake of Chemicals from the Soil

A vast number of studies on the uptake of xenobiotics showed that by far, most of these substances can smoothly diffuse across membranes [12,13,14]. The ability for such passive membrane transfer can roughly be deduced from the so-called *K*_OW_ value [10], which corresponds to the distribution coefficient of a certain substance between octanol and water. In most related publications, its decadal logarithm, denoted as p*K*_OW_, is outlined [33]. Frequently, this parameter is also denominated as log*P* [34]. Comprehensive research displayed that substances exhibiting log*P* values between −1 and 3 are inherently able to diffuse easily through biomembranes [33,35]. It is self-evident that these coherences, which pertain to the uptake of xenobiotics, also apply to all other organic substances. This phenomenon was approved by Hurtado et al. [36], who demonstrated that organic compounds exhibiting appropriate log*P* values are generally taken up by plants. In consequence, there is no doubt that also all plant-derived natural products exhibiting an appropriate log*P* value are passively taken up by plants when they are present in the soil.

Nevertheless, an additional factor must be considered when contemplating the membrane permeability of alkaloids, i.e., the pH-dependent protonation of these alkaline specialized metabolites [37]. Whereas the free bases of most alkaloids pass biomembranes smoothly, the protonated forms cannot, since the positive charge strongly increases the hydrophilic character and thus prevents the penetration of biomembranes. This manifestation nicely coincides with the quite negative log*P* values of charged alkaloid salts. This implies that—apart from the log*P* of the various alkaloids—the pH of the medium also strongly determines their membrane permeability and thereby massively influences the extent of the uptake of alkaloids from the soil [38,39]. In acidic soils, due to the high degree of protonation, most alkaloids are unable to cross the plasmalemma of the root cells and accordingly, they are not taken up [10]. In contrast, in soils exhibiting enhanced pH values, protonation is pushed back and the alkaloids exist as free bases, which smoothly penetrate the plasmalemma of the root cells. provoking their strong uptake [39].

### 2.2. Horizontal Natural Product Transfer between Vital Plants

When contemplating the basic coherences of the horizontal natural product transfer, the question arises whether—in analogy to the relocation of natural products from rotting plant material—a corresponding transfer also might occur between living plants. This seems to be obvious because it is well known that allelochemicals are also released into the environment by various mechanisms [40]: they might either be leached out of decomposing plant materials, or they could actively be exuded from vital plants by their roots [2,3] or by their leaves [40,41]. In order to ascertain a putative release of common and typical natural products into the soil and a subsequent transfer into acceptor plants, co-culture experiments have been conducted: *Senecio jacobaea* plants containing high concentrations of pyrrolizidine alkaloids (Pas) were cultivated in the same pot together with parsley plants. After two months of co-cultivation, the plants were harvested and analyzed. Amazingly, in all parsley plants, which had been co-cultivated in the same pot together with a *Senecio* plant, significant concentrations of Pas were present; on average more than 200 µg/kg d.w. [22]. To rule out that the transfer of alkaloids is due to a too intensive contact between the roots of the donor and acceptor plants, a further study employing various distances between the plants was required. Accordingly, in a corresponding field trial, different acceptor plants were co-cultivated with *S. jacobaea*. It turned out that, in all plants grown in the field together with *S. jacobaea*, significant amounts of PAs were also present [22]. Thus, the alkaloids, which previously had been synthesized in the donor plants, are now present in the plants growing in the vicinity. Accordingly, there is no doubt that the PAs indeed had been transferred from vital donor plants via the soil into acceptor plants growing nearby (Figure 1). In further approaches, it could be verified that non-alkaloidal specialized metabolites are also transferred via the soil from donor to acceptor plants, i.e., coumarins [28], stilbenes [29], aristolochic acids [42,43], or betalains [25]. Meanwhile, several studies outlined a corresponding transfer of specialized metabolites from vital donor plants to acceptor plants growing in their vicinity (Table 1). However, up to now, there is no clue concerning the mode of this transfer.

First and foremost, it could be assumed that the described PA-transfer between living plants is due to the shedding of *Senecio* leaves; when the leaves decay, the Pas could be leached out from the rotting plant materials. In principle, this possibility cannot be excluded, but during the pot experiments described above, no leaves had been shed [26]. Thus, at least in these experiments, a transfer via shed leaves could be ruled out. Nonetheless, when referring to the mentioned field experiment, such an option cannot be excluded. These coherences also account for most of the reports on the transfer of specialized metabolites mentioned in Table 1. When considering an entire vegetation period, it has to be regarded that a senescence-induced shedding always takes place. Thus, this option certainly might be relevant, at least to some extent.

Another option for the transfer of the alkaloids could be due to their bleeding out from minor leaf injuries, which might be caused by herbivore or pathogen attacks. Yet, the plants used in the co-culture experiments had been described as healthy and no marks of herbivory were visible [22]. Thus, it seems unlikely that the observed PA-transfer is related to plain bleeding of the alkaloids from injured leaves. This conclusion is underlined by the massive differences in the PA-spectra of donor and acceptor plants [22]. In case of wound-induced bleeding, all the different PAs accumulated in the donor plant would be leached out and taken up by the acceptor plants to more or less the same extent. The observed differences must be due to a process, whose extent differs for the various alkaloids. In consequence, simple bleeding of the alkaloids from injured leaves can be ruled out. The same deduction is also valid for wound-induced bleeding from putatively injured roots. Thus, we have to suppose that the PAs present in the acceptor plants indeed had been released from vital, healthy and uninjured *Senecio* donor plants.

When bearing in mind the well-established active exudation of allelochemicals [2,3,40], it seems to be obvious that PAs, as well as other alkaloids, are also exuded by the donor plants, either by their roots or by their leaves. When discussing such a release, it readily becomes apparent that in principle there are two possibilities: the alkaloids could either be diffused passively out of the cells, or they could actively be exuded by the roots or shoots. In this context, the finding of García-Jorgensen et al. [47] becomes relevant, which outlined that the carcinogenic phytotoxin ptaquiloside, which is accumulated in the fern *Pteridium aquilinum*, indeed passively diffuses out of its leaves. Consequently, all alkaloids and other specialized metabolites exhibiting appropriate log*P* values might also simply diffuse passively out of leaves or roots into the soil, and accordingly will be taken up passively by the roots of plants growing in the vicinity. Yet, in case of polyphenolic compounds, an uptake might be prevented due to their oxidation to yield quinones, which efficiently bind to proteins, forming insoluble tanning products. The observed differences in the PA spectra of donor and acceptor plants might result from disparities in the log*P*- and p*K*_a_-values of the various PAs (see below) and thus, in their ability to diffuse through biomembranes. However, in this context, it has to be considered that in plant cells alkaloids are efficiently trapped in their vacuoles. Already half a century ago, Matile [48] outlined his famous “ion trap mechanism”: within the neutral cytosol, most alkaloids are present as free bases and—provided that their log*P*-values are in the range between -1 and 3 (see above), they are able to diffuse through biomembranes, such as the tonoplast. However, due to the acidic conditions in the vacuoles, the alkaloids become protonated. Consequently, they are no longer able to pass the tonoplast and accordingly are trapped and accumulated in the vacuoles. Thus, a passive release out of the cells is quite unlikely. Yet, these coherences only apply to alkaloids revealing a relatively high p*K*a, ensuring a sufficient protonation in the vacuole [39]. In contrast, alkaloids, which reveal a very low p*K*a, e.g., caffeine (p*K*a = 0.7) are not protonated in the acidic vacuole. Since they do not exhibit a positive charge that typically prevents their membrane permeability, these alkaloids are not retained within this compartment. Nevertheless, due to an effective complex formation with chlorogenic acids, caffeine—to a certain extent—is also retained in the vacuole [49], since—in the same manner as protonated alkaloids—the caffeine-chlorogenic acids-complex also cannot pass the tonoplast passively and thus, it is retained in the vacuole.

Most of the literature dealing with the release of alkaloids from vital plant parts is based on organ or cell culture experiments, e.g., harmine and harmaline are exuded from root culture cells of *Oxalis tuberose* [50], the indole alkaloid ajmalicine is secreted from hairy root cultures of *Catharanthus roseus* [51], and nicotine is present in the medium of root cultures from *Nicotiana tabacum* [52]. Based on the coherences outlined above, the occurrence of alkaloids in the culture medium might just be a hint for an active exudation. Nonetheless, Toppel et al. [53] demonstrated that in the culture medium of *Senecio* root cells, senkirkine was exclusively present, although the cultured cells contained a wide variety of PAs. This discrepancy can only be explained by the fact that senkirkine is actively and specifically exuded from the cells into the culture medium [53]. It is evident that such exudation requires specific carriers, which catalyze the transfer of the protonated alkaloids across the tonoplast and thus out of the vacuole. Indeed, numerous alkaloid transporters are described in the literature [54,55,56], for review see [57]. Unfortunately, in most of these reports, the ability of alkaloids to easily pass biomembranes is entirely ignored, and the authors are not aware that the transporters are required for the membrane transfer of protonated alkaloids rather than for the free bases [27].

Apart from the data on the release of alkaloids into the medium of organ and cell cultures, various data on the liberation of alkaloids from genuine plants into the soil are available. Baumann and Gabriel [58] showed that the roots of coffee seedlings exude caffeine, and Schulz et al. [59] demonstrated that dihydroxybenzoxazinone is released from the roots of *Agropyron repens* into the soil. Analogously, *Oxalis tuberosa* roots liberate carboline alkaloids, i.e., harmine and harmaline, into the soil [60]. Moreover, lupine alkaloids (quinolizidine alkaloids) had been detected in soils well-vegetated by narrow-leaf and yellow lupines [61]. Similarly, Hama and Strobel [62] found PAs in soils, overgrown with PA-containing *Senecio* plants. Apart from the occurrence of various alkaloids in soils, the assumption that alkaloids are released by the roots of various alkaloidal plants is supported by various indirect results, i.e., by detecting them in acceptor plants growing in their vicinity (Table 1). However, when contemplating the mode of release of these alkaloids, any solid proof on the mechanism of the release is lacking. In principle, these alkaloids might be eluted from shed leaves or leached out from injured leaves or roots, respectively. Moreover, they might be released from vital tissues, either by passive diffusion or an active exudation by the means of transporters. Indeed, with respect to the data mentioned above, up to now, no option can ultimately be excluded. Much more research is required to soundly unveil the actual path of alkaloid release into the soil.

### 2.3. Translocation of Chemicals within the Acceptor Plants

The alkaloids, which had been taken up by the roots of the acceptor plants, are subsequently present in their leaves [19,20]. Thus, the imported specialized metabolites have to be translocated within the acceptor plants. In principle, there are only two options for such allocation, either via xylem or phloem. In the case of xenobiotics, it is well documented that their transport from root to shoot is accomplished via the xylem [33,63]. Accordingly, it seems to be obvious that analogously also the natural products taken up by acceptor plants are translocated within the xylem. This theory was confirmed by the finding that imported alkaloids accumulated to a high extent in old leaves, whereas the alkaloid concentrations in young leaves were quite lower [19,20]. This different distribution is due to expansion-related differences in the transpiration rates, which in turn drives the xylem transport, and thus is responsible for the different extent of allocation of substances via the xylem. Furthermore, no imported alkaloids had been detected in the flowers of the acceptor plants [20], which represent a major physiological sink. Thus, a source-sink-translocation via phloem can be excluded. Finally, the implication of a xylem-based allocation of imported natural products is verified by the finding that, within the acceptor plants, these substances are also present in guttation droplets [20], whose composition corresponds to that of the xylem sap [64]. From these coherences, it has to be deduced that the imported natural compounds, in particular nicotine and PAs, indeed are translocated within the acceptor plants in the xylem.

When comparing these insights with the transport of PAs in genuine alkaloidal plants, such as *Senecio*, a strong discrepancy becomes evident: In the group of Hartmann, it was soundly verified that the PAs synthesized in the roots of *Senecio* plants are transferred as PA-*N*-oxides via phloem into the physiological sinks, such as flowers and seeds [65,66,67]. Based on the physicochemical properties of the alkaloids and their corresponding *N*-oxides, this putative contradiction could be elucidated [37]: as outlined above, the free bases of alkaloids readily diffuse through biomembranes. Accordingly, due to the ion trap mechanism already explained for the accumulation of protonated alkaloids within the vacuoles (see above), they are also displaced from the alkaline phloem and trapped within the acidic xylem. In consequence, they are allocated within the xylem into the transpiring leaves—a fact that is fully in accordance with the findings displayed for the alkaloids taken up from the soil. In contrast, the strongly enhanced hydrophilic properties of alkaloid-*N*-oxides or quaternary alkaloids, respectively, are not able to diffuse through biomembranes. Consequently, they would be retained in the alkaline phloem, and thereby enable the option for a long-distance transport to the physiological sinks via phloem as reported by the Hartmann group [65,66,67].

Since the specialized metabolites are passively taken up by the roots of the acceptor plants, the velocity and the extent of their import should be quite similar in the various species. The only factor which determines the uptake is the corresponding difference in concentration between the soil and the cytosol of the rhizodermis cells. Nevertheless, this gradient between “inside and outside” depends on various additional factors. Apart from the pH, which strongly influences the share of protonated and thus membrane-permeable alkaloids (see above), the most important factor corresponds to the decrease in the concentration of imported compounds within the root cells and thus, how the concentration gradient between “inside and outside” is maintained. In principle, there are two possibilities, either their allocation into other organs or their modification (see below). The velocity of a xylem-based allocation is mainly determined by the rate of transpiration [68]. Indeed, this property varies greatly among different plant species and is therefore responsible for the observed differences in the uptake of a certain natural substance imported into different plants.

Additionally, the gradient of a certain substance between “inside and outside” might also be impacted by a corresponding decrease in its concentration in the soil, either by its adsorption to various soil particles or by a related microbial decomposition or modification, respectively.

In addition to these insights, the transfer of natural products from living donor plants might also open new doors to unveil the beneficial impact of some hitherto not fully understood processes, i.e., co-cultivation of certain vegetables and crop rotations. The findings related to the exchange of natural products among vital plants might provide new approaches for the explanations of these ambiguous phenomena.

### 2.4. Modification of the Imported Compounds in the Acceptor Plants

Astonishingly, the contents of alkaloids imported into the acceptor plants decreased over time: a massive decline in concentration was reported for the content of nicotine in peppermint plants [19], as well as for the PAs in various spice plants [20]. Since a complete decomposition of these compounds to CO_2_ and water seemed to be very unlikely, it was argued that the imported compounds had been modified within the acceptor plants. In this context, it is appropriate to outline that xenobiotics, which are taken up from the soil, are also known to be frequently modified within the acceptor plants (Figure 1). The most abundant modifications comprise oxidations, hydroxylations, and conjugations with glucose [69,70]. Based on these cognitions, the so-called green liver concept—a hypothesis that proposed that these reactions are part of a deliberate detoxification system for xenobiotics—was developed [70,71]. In consequence, it seems to be natural that the time-dependent decrease in the alkaloid content in the acceptor plants might also be due to the modification of the imported natural compounds. The confirmation of this conjecture was elaborated by advanced analyses of PAs present in the acceptor plants: standard quantification of PAs is generally performed based on standard LC-MS methods, quantifying and summing up the individual contents of 27 genuine PAs [20,72]. Thus, in these studies, solely the original already known PAs, which had previously been present in the donor plant, were estimated, whereas their putative derivatives could not be determined. Yet, in a further approach to quantify the PAs in the acceptor plants, an alternative method—denoted as “sum parameter method”—was employed. For this approach, the PAs are degraded to their basic skeleton, the so-called necine base, which subsequently is quantified by HPLC-ESI-MS [73]. In consequence, apart from the genuine Pas, all putative derivatives will also be determined. The related results were really astonishing: in contrast to the putative decline of the alkaloids over time—as estimated by applying the standard method—a further massive increase in the alkaloids content was verified [30]. In parsley, two weeks after the mulching with PA-containing *Senecio* plant material, more than three-quarters of the imported alkaloids had already been modified within the acceptor plants [30], and thus became invisible to the standard LC-MS quantification. Unfortunately, up to now, the structure of these PA derivatives could not be elucidated. There is a massive demand for further research, since the content of alkaloids in contaminated plant-derived commodities is obviously far higher than that previously stated. This especially accounts for PAs to soundly evaluate the actual health risk.

In order to elucidate the basics of the modification of imported natural products in the acceptor plants, fundamental studies employing umbelliferone as a model substance had been performed. In contrast to most alkaloids, due to their high fluorescence, coumarins as well as their putative derivatives could easily be identified. The studies dealing with the uptake of umbelliferone into various acceptor plants clearly unveiled that the metabolic fate of imported natural products strongly depends on the plant species [28]: in various acceptor plants like flax, radish, or pea, the umbelliferone imported by their roots is just accumulated in their leaves. In contrast, in garden cress and barley, it is hydroxylated to yield esculetin. However, in garden cress, the generated esculetin is glucosylated to esculin, whereas in barley, it is methylated to yield scopoletin [28]. Based on the employment of naproxen, an efficient inhibitor of P450 enzymes, it was demonstrated that the oxidative conversion of umbelliferone is catalyzed by cytochrome P450 enzymes [74]. In contrast, the oxidation of harmaline taken up by barley seedlings to yield harmine is not catalyzed by a P450 enzyme [75].

From the coherences mentioned above—in particular that the extent and mode of modification of the imported products strongly vary between plant species—the occurrence of a general detoxification system, hypothesized as “green liver concept” [69,71], has to be doubted. In contrast, the imported substances seem to be accidentally modified by enzymes involved in the inherent specialized metabolism, which are present in some plants and absent in others [27].

## 3. Horizontal Natural Product Transfer and Allelopathy

Allelochemicals released from numerous plants inhibit potential competitors for limited light, water, or nutrient resources [76]. For a number of these allelopathic compounds, their underlying physiological mechanism is related to the inhibition of a wide array of corresponding metabolic processes: allelochemicals can affect photosynthesis by reducing PSII efficiency, might impair nutrient uptake and ATP synthesis, or harm cell cycle and gene expression or the phytohormone metabolism [1,77]. Due to these inhibitory effects, they are already employed in agricultural systems as useful tools for controlling weeds [78,79], inhibiting pests [80,81], and fighting diseases [2,82]. In consequence, the usage of synthetical and potentially toxic plant-protecting agents can significantly be reduced. Indeed, the absorption of some allelochemicals by crop plants might also cause health risks as mentioned for the uptake of PAs. However, in general, the related concentrations are far too low to entail serious consequences for human health.

Apart from that, many plants are able to cope with inhibitory allelochemicals. Whereas seed germination of cucumber, alfalfa, garden cress, or tomato is strongly inhibited by juglone, the germination of seeds from wheat, barley, watermelon, corn, and radish is not affected; and the seedling growth of muskmelon is even enhanced by juglone [83]. Obviously, the sensitivity of seeds for allelochemicals varies strongly between different plant species. As outlined above, the corresponding mechanisms seem to be quite diverse and may pertain to various levels of metabolic processes. In this sense, several primary responses to the presence of allelochemicals are described. One of the most important mechanisms of the inhibitory effect of juglone is due to the generation of reactive oxygen species (ROS) [84]. However, the application of juglone may also enhance the ROS-scavenging systems and thereby reduce radical-related damages [84]. Furthermore, detoxifying enzymes and transporters, which inactivate or eliminate the toxic compounds, can be induced by the allelochemicals. In maize, a glutathione transferase, which modifies juglone, is up-regulated in response to the presence of the allelochemical. As a consequence of this derivatization, its toxic properties are strongly diminished [85]. A comprehensive study on the derivatization of allelochemicals by Schulz et al. [59] outlined that the mode and extent of the detoxification process may vary between different plant species. The authors showed that many acceptor plants detoxify benzoxazinones, such as DIBOA and DIMBO, by glucosylation, whereas others detoxify them by the generation of related carbamates. Furthermore, in several other plant species, some further metabolites of the benzoxazinones had been detected [86]. Apart from varying sensitivities of acceptor plants for allelochemicals based on differences in the detoxification mechanism, the different sensitivities might also be due to quite other causations. In this sense, the actual inhibitory effects of a certain allelochemical could be more or less pronounced in different plant species. In this sense, Ding et al. [87], unveiled the molecular mechanism, being responsible for the massive differences in the toxicity of cinnamic acid, which is released as an allelochemical from cucumber roots. Even in the roots of *Cucumis sativus*, the cinnamic acid significantly decreases the activity of membrane H^+^-ATPase, whereas no corresponding inhibition is observed in the roots of fig-leaf gourd (*Cucurbita ficifolia*). Accordingly, it can be argued that observed differences in the sensitivity for a certain allelochemical are indeed related to differences in its direct impact on metabolic events.

These examples outline that more than one factor or mode of action might be responsible for the observed differences in sensitivity. These coherences become even more complex when considering that further environmental factors, e.g., stress, could alter the sensitivity for a certain allelochemical. In this context, the observation of Amari et al. [88] became evident, who showed that the sensitivity for juglone is strongly diminished when reactions involved in the SAR (systemic acquired resistance) are induced. Furthermore, it has to be considered that allelopathy comprises a wide array of interaction between plants by influencing their competition either by the direct effects of allelochemicals, e.g., by inhibiting the germination and growth of potential competitors or by indirect impacts, e.g., by affecting the microbiome, the nutrient availability, and the character of the soil [89].

At this juncture, the question arises whether the observed differences in the sensitivity of the various plant species indeed originate unfailingly from evolutionary adaptations or if some of them are due to random and accidental detoxification processes catalyzed by intrinsic promiscuous enzymes. Depending on the plant species, the specialized metabolism and thus, the composition of immanent enzymes, massively differs. In consequence—in analogy to the proposed random modification of xenobiotics [27]—in some plants the imported allelochemicals might also be modified “accidentally”, whereas in others, they are not. However, in contrast to allelopathic interactions which represent quite ancient features, plants had been confronted with xenobiotics, i.e., herbicides or fungicides only for one and a half century. The timespan until today is far too low for the evolutionary generation of a comprehensive detoxification system, which is aimed to “deliberately” detoxify these xenobiotics, i.e., as proposed by the so-called green liver concept [69,71]. Nonetheless, the processes generating the well-known resistances against herbicide can be realized in a very short time, since only one or few mutations are necessary to thwart the toxic effect, i.e., mutations in genes encoding enzyme proteins, non-enzyme proteins, or factors controlling cell division [90]. In consequence, it could be argued that the postulated green liver concept for the detoxification of xenobiotics [69,71] does not represent an adaptation of plants coping with synthetic herbicides or fungicides, but that these detoxification systems had obviously evolved in response to toxic allelochemicals. In consequence, the evolution of the various adaptations of plants to cope with allelochemicals of plants growing in their vicinity ought to be considered as important and principal factors for the deployment and sustaining of a certain plant community. Unfortunately, these insights have not been considered so far, either in plant sociology or in vegetation science.

## 4. Conclusions

The coherences outlined in this treatise display that the horizontal natural product transfer, which represents a quite general mechanism pertaining to all plants, might be the basis for both the evolution of allelochemicals and subsequently, of the corresponding detoxification mechanisms, which in turn currently also include coping with xenobiotics. Moreover, the insights into the differences in the potential detoxification of allelochemicals may open new doors for the understanding of “why and how” the composition of the various plant communities has been evolved and preserved.

## Figures and Tables

**Figure 1 plants-11-03264-f001:**
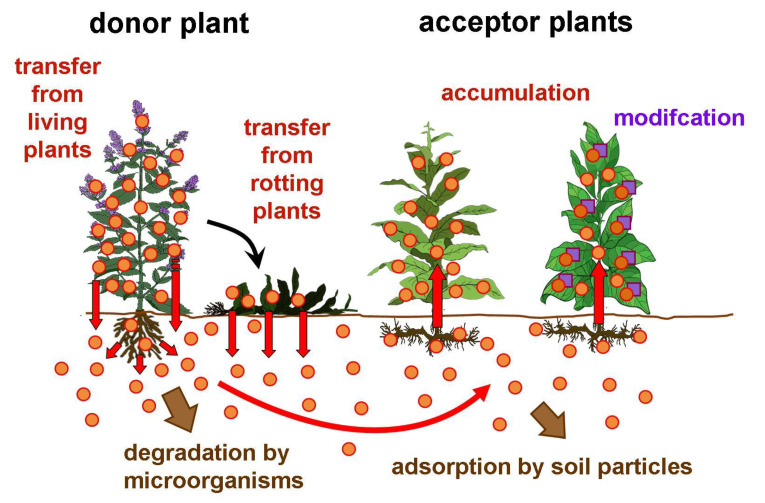
Horizontal natural product transfer. The scheme displayed by Selmar et al. [26] is supplemented with the processes occurring in the rhizosphere.

**Table 1 plants-11-03264-t001:** Transfer of alkaloids via soil between vital donor and acceptor plants.

Donor Plant	Type of Alkaloid	Acceptor Plant	Authors
*Chromolaena odorata*	Pyrrolizidine alkaloids	*Zea mays*	[44]
*Secale cereale*	Benzoxazinoids	*Vicia villosa*	[45]
*Senecio jacobaea*	Pyrrolizidine alkaloids	*Matricaria chamomilla*	[20,22]
*Senecio jacobaea*	Pyrrolizidine alkaloids	*Melissa officinalis*	[20,22]
*Senecio jacobaea*	Pyrrolizidine alkaloids	*Mentha × piperita*	[20,22]
*Senecio jacobaea*	Pyrrolizidine alkaloids	*Petroselinum crispum*	[20,22,25]
*Senecio jacobaea*	Pyrrolizidine alkaloids	*Tropaeolum majus*	[22]
*Solanaceae*	Atropine	*Triticum aestivum*	[46]

## Data Availability

Not applicable.

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
