# Peer review of "Novel Cognitions in Allelopathy: Implications from the “Horizontal Natural Product Transfer”"

_plants, 2022, doi:10.3390/plants11233264_

Round 1

Reviewer 1 Report

The authors did a good work from an experimental point of view, and I recommend the article for publication after some minor revisions.

More specific:

L188: Can this phenomenon occur in general for polyphenols?

Please insert Table 1 and Figure 1 into your main manuscript.

In Figure 1 correct the word ‘’modification’’.

Author Response

L188: Can this phenomenon occur in general for polyphenols?

We introduced in the text:   … and accordingly will be taken up passively by the roots of plants growing in the vicinity. Yet, in case of polyphenolic compounds, an uptake might be prevented due to their oxidation to yield quinones, which efficiently bind to proteins forming insoluble tanning products. The observed differences in the PA spectra of donor and acceptor plants might result from disparities in the logP- and pKa-values of the various PAs (see below) and thus, in their ability to diffuse through biomembranes.. However

Please insert Table 1 and Figure 1 into your main manuscript.

 - done-

In Figure 1 correct the word ‘’modification’’.

 - done -

Reviewer 2 Report

The authors have written several reviews in this topic in the last few years (among others: 2015: 10.4172/2161-0525.1000287; 2017: 10.1007/978-3-319-68717-9_12; 2019: 10.1021/acs.jafc.9b03619; 2020: 10.1007/978-3-319-96397-6_10). The recent review, submitted to Plants, hardly contains novelty when it is compared to the former ones. Only the 3rd chapter (“Horizontal Natural Product Transfer and Allelopathy”) could be considered as something “new”, however, its content is also known and reviewed. The authors’ last relevant review was published in 2020 (see above). So, it would be a viewpoint to review the literature and the new data and concepts published since that. Despite this, 91% of the references originate before 2020. The one and only figure is practically the same as the one published several times. I do not think this manuscript could provide novel information sufficient for a new and independent review.

Author Response

Indeed, most statements of the reviewer are correct; I already mentioned this problem explicitly when I answered the request to contribute to this “Special Issue”. As agreed, I avoided unnecessary replications and redundancies and focussed in the actual review on various allelopathic aspects. I am convinced that this endeavour was successful and our ms. provides a valuable contribution to the “Special issue”, and accordingly, the caveat of the reviewer is rebutted.

Reviewer 3 Report

horizontal transfer of allelochemicals and its relevance for allelopathic interactions. These knowledges of specialized metabolites and their translocation between plants should significantly contribute to understanding of plant-plant interactions, and to the evolution of typical effects, such as inhibition of growth and germination of potential competitors. The authors have analysed numerous recent references of plants metabolites as allelochemicals and their uptake and transfers to other plants, uptake of chemicals from the soil, translocation of chemicals within the acceptor plants, modification of the imported compounds in the acceptor plants, and effects of allelochemicals. This material will be useful for various concerned researchers and specialists. Therefore, the article might be recommended for publication in the special issue of the Journal. 

Author Response

- no action required -

Round 2

Reviewer 2 Report

I am terribly sorry but this is practically the same review, so, my opinion also remains the same.

Author Response

Dear editor,

as I have already explained in my first response to this reviewer, I mentioned this problem, when I answered your request to contribute to this “Special Issue”. Regardless to this issue, you agreed; and in the revised ms., I avoided unnecessary replications and redundancies and focussed in the actual review on various allelopathic aspects. I am really convinced that this endeavour was successful and our ms. provides a valuable contribution to the “Special issue”, and accordingly, the caveat of the reviewer is rebutted.

Greetings from Braunschweig

Dirk Selmar